# MicroAnnot: A Dedicated Workflow for Accurate Microsporidian Genome Annotation

**DOI:** 10.3390/ijms25020880

**Published:** 2024-01-10

**Authors:** Jérémy Tournayre, Valérie Polonais, Ivan Wawrzyniak, Reginald Florian Akossi, Nicolas Parisot, Emmanuelle Lerat, Frédéric Delbac, Pierre Souvignet, Matthieu Reichstadt, Eric Peyretaillade

**Affiliations:** 1INRAE, UMR Herbivores, Université Clermont Auvergne, VetAgro Sup, 63122 Saint-Genès-Champanelle, France; jeremy.tournayre@inrae.fr (J.T.); pierre.souvignet@inrae.fr (P.S.); matthieu.reichstadt@inrae.fr (M.R.); 2LMGE, CNRS, Université Clermont Auvergne, 63000 Clermont-Ferrand, France; valerie.polonais@uca.fr (V.P.); ivan.wawrzyniak@uca.fr (I.W.); reginald.f.akossi@gmail.fr (R.F.A.); frederic.delbac@uca.fr (F.D.); 3UMR 203, BF2I, INRAE, INSA Lyon, Université de Lyon, 69621 Villeurbanne, France; 4VAS, CNRS, UMR5558, LBBE, Université Claude Bernard Lyon 1, 69622 Villeurbanne, France; emmanuelle.lerat@univ-lyon1.fr

**Keywords:** microsporidia, structural annotation, dedicated workflow, high quality annotation

## Abstract

With nearly 1700 species, Microsporidia represent a group of obligate intracellular eukaryotes with veterinary, economic and medical impacts. To help understand the biological functions of these microorganisms, complete genome sequencing is routinely used. Nevertheless, the proper prediction of their gene catalogue is challenging due to their taxon-specific evolutionary features. As innovative genome annotation strategies are needed to obtain a representative snapshot of the overall lifestyle of these parasites, the MicroAnnot tool, a dedicated workflow for microsporidian sequence annotation using data from curated databases of accurately annotated microsporidian genes, has been developed. Furthermore, specific modules have been implemented to perform small gene (<300 bp) and transposable element identification. Finally, functional annotation was performed using the signature-based InterProScan software. MicroAnnot’s accuracy has been verified by the re-annotation of four microsporidian genomes for which structural annotation had previously been validated. With its comparative approach and transcriptional signal identification method, MicroAnnot provides an accurate prediction of translation initiation sites, an efficient identification of transposable elements, as well as high specificity and sensitivity for microsporidian genes, including those under 300 bp.

## 1. Introduction

The microsporidian phylum, an evolved branch of the rozellids [1], includes over 1700 species divided into more than 220 genera infecting an extremely diverse range of hosts protists to all major animal phyla [2]. Microsporidia can be highly represented in some aquatic environments, and thus play a role in food webs [3,4,5]. Numerous species can also be found, infecting animals of veterinary and economic importance with prevalence rates reaching 15% in bovines and 22% in ovis [4,5]. In addition, some microsporidian species may be involved in human diseases, with 17 species belonging to 10 genera that have been described as leading to severe syndromes, predominantly in immunocompromised patients [6]. This includes patients with acquired immunodeficiency syndrome (AIDS), in whom the prevalence of microsporidia can be as high as 11.2% [5], but also patients that have undergone organ transplants and been treated with immunosuppressive drugs, with a prevalence of up to 13.2% [5,6,7,8,9]. 

With the advent of next-generation sequencing (NGS) technologies, the systematic sequencing of microsporidian genomes has been undertaken and, over the last few decades, genomic data have rapidly been accumulating, with more than 50 genomes now available [10]. The study of these genomes has allowed researchers to highlight the consequences of the distinct evolutionary patterns in this parasitic group. Indeed, due to their adaptation to obligate intracellular parasitism, the genomes of Microsporidia are under strong selective pressures, which conduct them to their present specific characteristics [10,11,12]. Thus, microsporidian genome sizes are highly reduced, with some species having reduced their genomic content to potentially the lowest limit required for life [11]. The human-infecting species *Encephalitozoon intestinalis* with 2.3 Mbp harbours the smallest microsporidian genome that has been sequenced [13]. The genomic reduction in microsporidia is not just limited to a massive loss of genes; it also affects the gene length. For example, *Encephalitozoon cuniculi* Coding DNA Sequences (CDSs) are, on average, 15% shorter than their yeast orthologs [14]. This CDS size reduction also leads to around 8.5% of the CDSs having a size under 300 nucleotides [15,16]. Another consequence of microsporidian gene compaction is the removal of intronic sequences. Introns with reduced size may remain in a small number of genes; however, this is not common to all species, thus making their prediction more difficult [16,17]. Microsporidian genes also present a strong compaction of the 5′ and 3′UnTranslated Regions (UTRs). In some cases, the 5′UTR can even be absent, and the transcribed mRNAs begin with the translation initiation codon [15,18,19,20,21,22]. This high reduction in 5′UTR length seems to be an advantage for the identification of transcriptional regulation signals, which are therefore localized near the translation initiation codon. Numerous studies have shown that these signals are highly conserved amongst microsporidian species and consensus sequences have been defined [15,16,21]. Thus, CCC-like or GGG-like signals are located upstream of the Translation Initiation Site (TIS). In genomes with a low G+C content, these signals are often replaced by a strong A+T-bias (more than 90%) near the TIS. A final feature affecting the size of microsporidian genomes is the shortening of intergenic regions, which are essential for transcription as they contain promoters and enhancers [11].

Microsporidian genomes are characterized by a high rate of sequence evolution, which has induced difficulties in positioning these microorganisms in the phylogenetic eukaryotic tree [1]. The identification of orthologous genes between microsporidian species has also proved to be challenging, even for genes crucial to their infectious process [23] or DNA repair [24]. While genomes tend to present an extreme reduction, one exceptionally large microsporidian genome of 51 Mbp has been reported in the mosquito parasite *Edhazardia aedis*. This is mainly due to the expansion of Transposable Element (TE) families in the genome of this microsporidian species. The high rate of sequence evolution in the microsporidian phylum also concerns TEs, making them hard to identify using comparative approaches [25,26].

Rapid and cost-effective next-generation sequencing (NGS) technologies have produced, and are still producing, numerous new microsporidian genome sequences. After genome assembly, efficient genome annotation represents a crucial step in pointing out all the biological processes that govern the life of these microorganisms. However, the previously described microsporidian genome features turned out to be the pitfall of classical computational methods [27] aiming to produce an accurate prediction of complete gene repertoires. Until now, microsporidian genomes have been annotated using ab initio protein predictions that were primarily based on the detection of Open Reading Frames (ORF) using various generalist softwares, such as GeneMarkES, AUGUSTUS [28,29] or Glimmer, which could be combined with the detection of CCC- and GGG-like motifs found in close proximity to microsporidian TISs [16,30,31]. Such signals significantly improve the prediction of translational initiation codons, which are otherwise defined as the first AUG codon of the studied ORF. In addition, extrinsic data, such as those available from orthologous gene sequences, have also been intensely used to carry out structural annotation [32]. Due to the high rate of sequence evolution in the microsporidian phylum, such approaches require an optimization of the comparison tool parameters (e.g., with BLAST). These parameters also need to ensure as unambiguous a prediction of TEs as possible. Finally, small protein-coding genes are often overlooked during structural annotation due to their shortness, lack of sequence conservation, and/or lack of known functions [33].

To address the challenging question of microsporidian genome annotation, we developed a dedicated annotation pipeline called MicroAnnot. MicroAnnot ensures gene prediction, as well as structural and functional annotation. Firstly, using curated databases of validated proteomes from four microsporidian species, an extrinsic approach was implemented using the BLAST tools [34] with optimized parameters to identify divergent or small orthologous sequences. The results were also exploited to predict potential translation initiation codons that were further validated using upstream transcriptional signals. Secondly, this comparative approach does not allow for identification of all the genes, particularly those forming part of the microsporidian pan-genome; these newly predicted genes were used as a training set for the Glimmer tool [35]. Thus, an ab initio sequence annotation was carried out and the potential translation initiation codons for the newly identified sequences were validated again using transcriptional signals. All predicted CDSs were then used as queries against a microsporidian-specific TE database. The identification of rRNAs was also carried out via a comparative approach using a database containing small subunit (SSU) rRNA sequences that were representative of all the microsporidian phylogenetic diversity and tRNA detection achieved using tRNAScan-SE [36]. MicroAnnot was tested on four microsporidian genomes and yielded a higher quality annotation of all the evaluated criteria (specificity, sensitivity, translational initiation codon prediction, small gene characterization and TE identification). Furthermore, functional annotation using the InterProScan tool [37] can also be included in the result files in either GENBANK [38], EMBL [39] or GFF [40] formats. MicroAnnot is available to the community at https://microannot.org and its source code is available at https://github.com/JeremyTournayre/MicroAnnot.

## 2. Results

### 2.1. MicroAnnot Usage

By selecting “New analysis” in the right web banner (Figure 1A), the MicroAnnot tool will display different parameters that can be modulated (Figure 1B). The software takes a flat file containing microsporidian genome FASTA sequence(s) as an input to be annotated (maximum size data 100 Mbytes). The web interface provides the user with the possibility of adjusting the threshold values of the different approaches and softwares used. The user may also select the training dataset for the Glimmer software (version 3.02) when fewer than 50 CDS are identified through the comparative search approach. Functional annotation using the InterProScan software (version 5.60–92.0). is disabled by default but can be enabled by clicking on the “activate” box.

When the input file is uploaded, the analysis parameters defined, and the databases selected, the analysis can be launched by clicking on the “Submit” button. Once the job submission is completed, the user can follow its execution (Figure 2A,B). Once the job is finished, several compressed archives containing GENBANK, EMBL and GFF formats can be directly downloaded (Figure 2C). In addition, an archive is also generated containing uncertain annotations that must be manually validated (see algorithm description below).

### 2.2. MicroAnnot Algorithm

The details of the MicroAnnot algorithm are shown in Figure 3. Briefly, the complete set of Open Reading Frames (ORFs) of a given minimum size (default value 240 nt) is extracted from input FASTA sequence(s) and, after translation, is used as a query to perform an initial BLASTP analysis against well-annotated microsporidian proteomes incorporated in MicroAnnot, namely *E. cuniculi*, *Nosema ceranae*, *Enterocytozoon bieneusi* and *Anncaliia algerae* [16]. As some microsporidian genes exhibit a drastic size reduction, a second BLASTP analysis is performed after extracting all CDS sequences coding for short proteins (less than 80 amino acids, Figure 3 left part). For this second analysis, BLASTP parameters are optimized to identify sequence similarities between short peptide sequences and a specific database constructed from biologically and/or manually validated small gene sequences [15,16]. The significant results of the two BLASTP analyses are processed to optimize the identification of TISs, either by considering the alignment of the N-terminal part of the proteins or via the presence of transcriptional signals in the proximity of the start initiation codon when the N-terminal alignment is not available after local BLASTP alignment.

The results of the first BLASTP analysis are also exploited to identify potential (i) frameshifts, (ii) introns and (iii) 5′ truncated CDSs. If more than 50 CDSs are well predicted by our first homology approach, their sequences are used to train the Glimmer model. Otherwise, the user has the possibility of using one of the Glimmer models defined from the four well-predicted proteomes. The TISs predicted by Glimmer annotation are then validated by the identification of upstream transcriptional signals. If these signals are absent, the CDS sequence is scanned to highlight these signals upstream of each potential translation initiation codon. As soon as an ATG codon is closely preceded by these signals, it is considered the correct translation initiation codon and the predicted CDS is adjusted.

In parallel, MicroAnnot is also used to annotate non-coding features. This is achieved using dedicated methods and tools. Transfer RNA (tRNA) annotation is conducted using the tRNAscanSE tool [36] directly embedded in the MicroAnnot pipeline. For the identification of rRNA-encoding genes (16S-23S rRNA units), a BLASTN analysis is conducted using the user’s input sequences as a query against a rRNA personal database, including sequences representing the complete phylogenetical diversity of microsporidian rRNAs [41].

To avoid redundancy when compiling all generated annotation data, all Glimmer predicted CDSs overlapping with other predictions are eliminated. We consider homology-driven annotation to be more reliable than Glimmer annotations, and this step helps to keep only the best annotations in MicroAnnot’s final output. We set an exception for Glimmer CDSs above 500 nucleotides, which are retained if they have fewer than 50 nucleotides overlapping with other predictions.

After overlap curation, potential transposable element (TE) sequences are also identified by comparing all predicted CDSs to a microsporidian-specific TE database using a TBLASTX approach. Finally, a functional annotation of all translated CDSs predicted by MicroAnnot is carried out using InterProScan. This step is optional and deactivated by default (Figure 1B).

The final annotation results are available to the user in GENBANK, EMBL and GFF annotation formats. An additional text file called “warning” is also generated and contains all the annotations for which the MicroAnnot tool invites caution. This includes potential frameshifts, introns, 5′ truncated genes, TIS that are too far from the ORF start, and overlapping genes. In all these cases, the CDS feature created in the output files is replaced by the “gene” feature and a warning note is added.

### 2.3. MicroAnnot Validation

The annotation of *E. cuniculi*, *N. ceranae* and *E. bieneusi* genomes was carried out by using MicroAnnot to select only the proteomes that did not correspond to those of the genomes that were to be annotated. For *E. intestinalis*, all the MicroAnnot proteomes were used.

A comparative analysis of the annotations proposed by MicroAnnot and several previous annotations revealed a higher sensitivity and specificity for MicroAnnot in regard to manually curated reference annotations [16] (Table 1 and Table 2, and Appendix A). Indeed, more than 95% of the genes that could previously only be identified by manual annotation are predicted, with this percentage rising to about 99.4% for the *E. intestinalis* species. It should also be noted that the MicroAnnot tool is efficient for the annotation of new genes, including 296 for *N. ceranae*. MicroAnnot reported some false positives, but their number remained relatively low, with specificity values between 98.4% and 99.9%. Furthermore, for the species harbouring the highest numbers of mispredicted genes, a large number contain a warning (27/39) to invite the user to manually control these predictions for *N. ceranae* species, and for the *E. bieneusi* species, 18 out of 25 falsely predicted genes correspond to low-complexity sequences misidentified by the Glimmer tool (Appendix A). For genes that were initially mispredicted within these genomes, the MicroAnnot tool also produced a significant improvement, with less than 25% false positives overall, and none of the genes incorrectly predicted in previous annotations of the *E. cuniculi* and *E. intestinalis* genomes being detected. MicroAnnot’s algorithm can also identify introns, frameshifts, sequencing errors and pseudogenes in sequences, and the comparative analysis reveals an efficiency of over 63% (*E. intestinalis*) going up to 100% (*E. cuniculi*) for intron identification, and between 39.5% (*E. bieneusi*) and 90% (*E. intestinalis*) for the detection of other non-canonical sequences and sequence features (frameshifts, sequencing errors, pseudogenes). For the determination of TISs, the MicroAnnot tool also displays conclusive results. Compared with the four other genome annotations and mispredictions, MicroAnnot predicts the correct translation initiation codon in over 70% of cases. Furthermore, additional TISs were corrected for the four species. Finally, the TE prediction module of MicroAnnot proved particularly effective, enabling us to show that 693 CDSs correspond to such elements in *N. ceranae*.

To validate MicroAnnot’s relevance and robustness, an analysis was carried out using the Funannotate tool [42,43], a pipeline including numerous software dedicated to ensuring gene prediction in both fungi and eukaryote genomes. Although this tool presents high specificity, with values close to those obtained with MicroAnnot, it is less effective in predicting the entire gene repertoire and corrected TIS of the studied (Table 2 and Appendix A).

## 3. Discussion

Although the correct annotation of genomes over the last decade has led to the development of increasingly innovative and specific approaches that consider the characteristics of each studied genome, this initial step in the in silico exploration of genomes is still often a source of error. Indeed, gene prediction is a complex process, especially in eukaryotes, and the prediction of all the genes in an organism is never 100% correct. A benchmark study of ab initio gene prediction methods in diverse eukaryotes using the most widely used gene prediction programs has shown that all these programs harbour numerous strengths but also various weaknesses [44]. These authors also concluded that ab initio gene structure prediction is a very challenging task, which should be further investigated [44]. For prokaryotic species, whose genome organization is close to that of Microsporidia, many common CDS predictors failed to identify the complete gene catalogue because some genes features fell outside the defined rules, such as non-standard codon usage, overlapping genes and small genes [45].

Due to their characteristics, the structural annotation of microsporidian genomes to define their genetic potential can quickly prove to be a real challenge. To take these constraints into consideration, we developed a tool dedicated to the annotation of these particular genomes. The annotation, carried out using the MicroAnnot tool on the four benchmark genomes, provides particularly conclusive results in terms of specificity and sensitivity, while conventional software used for the annotation of different microsporidian genomes (*E. cuniculi*; Glimmer prediction [14], *E. intestinalis*; BLAST procedures [13], *N. ceranae*; Glimmer [46] and *E. bieneusi*; FunGene and Glimmer3 [47]) do not offer predictions of the same quality given several badly predicted or even non-predicted genes, which can add up to as much as 10% of the total genes of the studied species [16].

Achieving complete genome annotation based on ab initio predictions can be particularly effective when it comes to model organisms, but the results in terms of sensitivity and specificity can drop for non-model species [44,48]. Annotations obtained using comparative methods, such as the alignment of protein sequences against predicted CDS, provide precise and reliable results [49]. However, due to the high rates of sequence evolution in microsporidian sequences [50], comparative approaches are difficult to implement with these species: the correct alignment of orthologous sequences from different microsporidian species and the parameters of the comparative tools all need to be optimised [10,51]. To provide high-quality alignments with the proteome sequences but also with the small gene sequences, the parameters of the BLAST software (version (2.13.0+), were modified, notably with the use of the BLOSUM45 matrix. This «deeper» matrix provides very sensitive similarity searches but also produces alignment overextension into less homologous regions, such as N-terminal regions [52]. The alignment of the N-terminal regions has been used by MicroAnnot to unambiguously determine gene TISs. Indeed, the correct recognition of this initiation codon is crucial in gene prediction to highlight the gene structure and its product [53]. Nevertheless, due to the high rate of sequence evolution, local alignment with BLAST software may stop before the methionine defining the N-terminal end of the sequence is reached. In this case, MicroAnnot considers the length of the homologous database sequence and the position of the BLAST local alignment to propose a potential translation initiation site in the query sequence. The validation of predicted TISs can be achieved by an evaluation of the ATG context [54]. The search for the Kozak sequence cannot be applied to microsporidian genes because the mRNAs are characterized by highly reduced or even absent 5′ untranslated regions (UTR), and only a bias in +4 position for an adenine or a guanine residue has been described [21]. However, 5′UTR size reduction represents an advantage because the transcription regulatory signals are in close proximity to the translation initiation codon. Characterized CCC-like or GGG-like signals, or a strong adenine/thymine-rich sequence (approximately 90%) upstream of TISs, are conserved within all microsporidian genomes [16,21,55,56], and their identification allows for the unambiguous support of TIS prediction. The search for these signals using the MicroAnnot tool proved particularly relevant by ensuring the correct prediction of more than 70% of the previously mispredicted TISs. In the case of *E. cuniculi*, more than 23% of TISs [15,16,21] were badly predicted during the first annotation of this genome without using the detection of such signals [14]. The annotation obtained using MicroAnnot, without considering the reference *Encephalitozoon* species, presents only around 20% of mispredicted TISs (Table 1). This value drops to less than 2.5% for the annotation of the *E. intestinalis* genome, carried out using the *E. cuniculi* proteome as a reference. It should also be noted that of the 32 badly predicted TISs in this species, 12 are contiguous to the putative correct ones (Appendix A). This comparative approach, coupled with the identification of the correct translation initiation codon, also proves relevant for identifying potential short introns that are mainly found within genes coding for ribosomal proteins [17,21,57].

The CCC- and GGG-like signals have been successfully used for the annotation of gene TISs in different microsporidian species such as *Ordospora colligata* [56] or *N. ceranae* [31]. These signals also proved relevant to ensuring the characterization of small microsporidian genes (CDS size < 300 nt), which are relatively frequent due to the general reduction in CDS sizes in microsporidia when compared to their orthologs in other fungal species [14]. Despite their number, these small genes are often misreported by generalist gene predictors. Advances in high-throughput technologies have highlighted an emerging world of proteins composed of small open reading frame-encoded micro-peptides [58]. Based on comparative approaches, the MicroAnnot tool proved particularly effective in ensuring the annotation of genes that had been ignored during the initial annotation of the four genomes studied in this work. These missing genes mostly correspond to small CDSs. For *N. ceranae*, 296 new genes have been identified, and 184 of them harbour a CDS smaller than 300 nt in length. Most of these new genes were previously highlighted by Pelin et al., who used an in-house script that combines Glimmer’s ab initio gene prediction algorithm, and CCC- and GGG-like motifs found in close proximity to microsporidian transcription initiation sites [31], reinforcing the relevance of our approach. Differences in genome sizes between microsporidian species are essentially due to the presence of TEs [25,59]. Unfortunately, gene predictors can predict CDS in these TEs, which can thus lead to an incorrect estimation of the number of genes in microsporidia [16]. The first step in structural annotation involving the exhaustive identification of repetitive elements is still challenging [27]. Despite their prevalence and importance, TE sequences remain poorly annotated and studied in almost all model systems [60]. The functional annotation of many microsporidian CDSs revealed that they contained specific TE ORF domains and motifs (see, for example, the product description of *Dictyocoela muelleri* species [61] in MicrosporidiaDB [62]. In addition, the identification of large multigene families in which some members have a low percentage of similarity with TE sequences shows difficulties in identifying certain TE families within microsporidian genomes [16]. Thus, MicroAnnot includes a specific module based on a comparative approach with well-predicted TEs to scan all predicted CDSs. This method allows for fine TE detection, and 693 predicted CDS were, in fact, TEs for the *N. ceranae* species.

Although structural annotation methods based on sequence homology are very effective, they are closely dependent on the presence of orthologous genes in the queried databases used for annotation. This is not a problem if we consider the pangenome, but it is more problematic for the core genome [63], especially for organisms such as microsporidia that can be found in multiple ecological niches and therefore present variable gene contents [26,64]. Therefore, to ensure the annotation of new genes, the implementation of an ab initio method is needed, and the Glimmer tool, which was used for the annotation of several microsporidian genomes, was selected [30,31,65,66]. For the best possible prediction, the dataset used for the construction of the Glimmer model must be perfectly reliable. The sequences included in the composition of this dataset are produced during the comparative annotation approach carried out by the MicroAnnot tool. This approach uses genes whose annotation was manually and, in some case, experimentally validated as a reference [15,16]. Hence, these sequences ensure the extraction of unambiguous CDSs with translational start sites that are well defined and validated by the presence of transcriptional regulatory signals in the upstream region. Once the CDSs are predicted by Glimmer, their position in the genome is evaluated, and this makes it possible to eliminate wrongly predicted genes using an overlap search. This overlap is notably responsible for the poor prediction of 20% and 28% of genes in *E. bieneusi* and *N. ceranae,* respectively [16]. Incorrectly predicted genes may also result from sequencing errors, leading to frameshifts. The comparative approaches utilized by MicroAnnot limit these erroneous gene predictions because the potential frameshifts are also evaluated. Frameshift characterization can only be carried out during the comparative approach step, and their identification is directly correlated to the reference proteomes available to the MicroAnnot tool, highlighting the importance of integrating additional reference genomes that are more representative of the phylogenetic diversity of microsporidia for better identification (see below). This type of error is, however, less frequent with improved sequencing approaches, base calling algorithms [67], and third-generation techniques [68,69]. The comparative analysis implemented in MicroAnnot also allows for the identification of pseudogenes that may be present in microsporidian genomes [70]. The MicroAnnot tool also mispredicted some genes. All these genes, however, were predicted during the ab initio annotation step with the Glimmer tool. This is likely linked to the Glimmer specificity concerns that were previously described [46]. However, for *N. ceranae*, 71% of incorrectly predicted genes (27 out of 38) harbour a “warning”, providing the possibility for the user to invalidate these predictions. As for the initial annotation of the *E. bieneusi* genome, sequences of low complexity are annotated as CDSs during the ab initio approach, but this number drops from 89 to 18 with the MicroAnnot tool, while these annotations are carried out with the Glimmer tool in both cases. 

To strengthen the relevance of the MicroAnnot, which considers the specific characteristics of microsporidian genomes, a comparative analysis was performed using the funannotate pipeline. This tool has a high sensitivity but presents limits in ensuring the prediction of the complete gene repertoire, especially small ones. The prediction of translation initiation codons is also less efficient. To improve these predictions, the tool can integrate RNA-seq data, and, in this case, proves to be particularly effective in the previous two points, but this requires RNA-seq data. The software also led to the prediction of numerous false introns due to the use of AUGUSTUS (version 3.4.0) and GeneMark-ES software (version 4.72), which have been optimized to ensure their prediction. As for the prediction of all genes, and the correct TIS, this tool also requires RNA-seq data, while the MicroAnnot tool does not.

Despite the progress made in developing increasingly efficient tools for genome annotation, the process requires manual curation to produce the most reliable results [27,71]. Furthermore, genome annotation is unfortunately not 100% accurate and needs to be regularly updated to take advantage of the new knowledge from comparative genomics, transcriptomics, proteomics, and metabolomics, which is continuously generated on the organisms under study, and more generally on all organisms, for annotation using sequence-similarity search approaches, for example. However, computer analysis methods lead to high levels of erroneous annotations which, when used, spread throughout international databases [72]. The MicroAnnot tool, while significantly increasing the sensitivity and specificity of predictions and reducing the number of incorrectly predicted TIS, is not yet 100% effective. For this reason, we plan to update it constantly, particularly regarding the content of the databases used for comparative approaches. Today, the tool includes four reference proteomes, but this number will need to be increased by implementing others available and validated proteomes, thus enabling the representation of all the microsporidian diversity. Meanwhile the sequencing of new microsporidian genomes, the annotation of genomes available in international databases and microsporidiaDB [62] for which transcriptomic data have also been produced to validate their annotation could rapidly be integrated in MicroAnnot for annotation via the comparative approach. The annotation of these genomes would also provide new sequences for the databases used by the software. Following the integration of a new reference genome, a check of the existing data should be systematically carried out, as this may enable errors to be corrected. The objective is not to propagate annotation errors, but to correct them over time by adding new sequences. The annotation of each microsporidian genome is, therefore, no longer fixed, but is a dynamic process enabling regular re-annotation [73].

Increasing the number of reference genomes would also be crucial for developing and integrating a specific module for the characterization of non-coding RNAs. High-throughput sequencing technologies such as RNA-seq have largely shed light on the world of ncRNA regulators [74], some of which have been systematically identified within microsporidian genomes using RNA-seq data [75,76,77,78]. Many methods predict ncRNA using sequence-derived features alone and they are difficult to apply to all species, especially microsporidia, which have a high rate of sequence evolution. To ensure the annotation of such ncRNAs, a synteny-driven “all-versus-all” BLASTN approach could be implemented following the addition of new genomes. This approach has previously been used to annotate the U1 small nuclear RNA [79], almost 15 years after the initial sequencing of the *E. cuniculi* genome [14].

## 4. Materials and Methods

### 4.1. Software Implementation

The software implementation utilized several tools and libraries for the analysis and processing of data. The following software components were employed: Perl (5.20.2), Bioperl (1.006924) with some modifications (a sort function was added on row 1264 in the genbank.pm file and on row 956 in the embl.pm file (usr/share/perl5/Bio/SeqIO); operator ‘eq’ was used instead of ‘==‘ on row 350 in the Simple.pm file (usr/share/perl5/Bio/Location)), dos2unix (6.0.4), ncbi-blast (2.13.0+), tRNAscan-SE (2.0.7), Glimmer (3.02), and InterProScan (5.60-92.0).

### 4.2. Web Interface

A web interface was developed to facilitate data access and analysis. The following technologies were used for the web interface implementation: PHP (7.4.26), MySQL (14.14), HTML, CSS and JavaScript with the libraries Bootstrap (4.3.1), datatables (1.10.2), font-awesome (6.0.0-beta2), swiper (8.4.4), Modernizr (2.8.3), jquery (v2.1.0) and nanoScrollerJS (0.8.0). Analyses were conducted on a Linux-based web server with the following specifications: Debian 3.16.7. 100 GB RAM Intel(R) Xeon(R) CPU E5-4620 0 @ 2.20 GHz. MicroAnnot source code is available at: https://github.com/JeremyTournayre/MicroAnnot.

### 4.3. Databases

To ensure comparative annotation, the gene product sequences of the *E. cuniculi*, *N. ceranae*, *A. algerae* and *E. bieneusi* genes for which manually curated annotation had been performed and whose curation approach had been experimentally validated by 5′RACE PCR [16] were integrated into the MicroAnnot tool.

To enable the identification of the small genes (CDS size less than 300 nucleotides) using the comparative approach, a specific in-house protein database was built from biologically and/or manually validated small gene sequences [15,16]. In addition, sequences in available microsporidian genomes that were orthologous to protein sequences of this database were added to the database. Some sequences annotated during the study of *N. ceranae* polyploidy [31] were also included in this database. Using all their respective CDS sequences, a Glimmer training dataset was built for each of the four reference genomes. To implement the TE database, all data from multiple published data sources were extracted [16,25,30,80]. These TE lists were completed by TE sequences identified thanks to the complete chromosome assembly of the *A. algerae* genome using PacBio Hifi sequencing technology. An exhaustive SSU rRNA database comprising the complete phylogenetical diversity of microsporidian rRNAs was built using data from [41].

### 4.4. MicroAnnot Analysis of the Four Microsporidian Genomes

To validate the MicroAnnot tool, sequences from four genomes for which annotation was manually carried out and published [16] were used as a query. They correspond to the sequences of *E. cuniculi* GB-M1 (GCA_000091225.2), *E. intestinalis* (GCA_000146465.1), *N. ceranae* BRL01 (GCA_000182985.1), and *E. bieneusi* H348 (GCA_000209485.1). For the *E. cuniculi*, *N. ceranae* and *E. bieneusi* genomes, annotations were carried out by selecting the four well-annotated microsporidian proteomes incorporated in MicroAnnot, with the exception of those corresponding to the genome used as a query. For the annotation of the *E. intestinalis* genome, all four proteomes were selected. The results of this comparative study are presented in Table 1 and Appendix A. For all other parameters, default values were used. In addition, the annotation of *E. intestinalis, N. ceranae* and *E. bieneusi* genomes was also performed with Funannotate pipeline (Galaxy Version 1.8.15) [42,43]. In this case, AUGUSTUS annotation was performed using the training species model of *E. cuniculi*. GeneMArk-ES was used in self-training mode. Finally, SNAP and GlimmerHMM tools were trained with the BUSCO microsporidia database, which encompasses 600 genes (microsporidia_odb10 dataset, creation date: 5 August 2020).

## Figures and Tables

**Figure 1 ijms-25-00880-f001:**
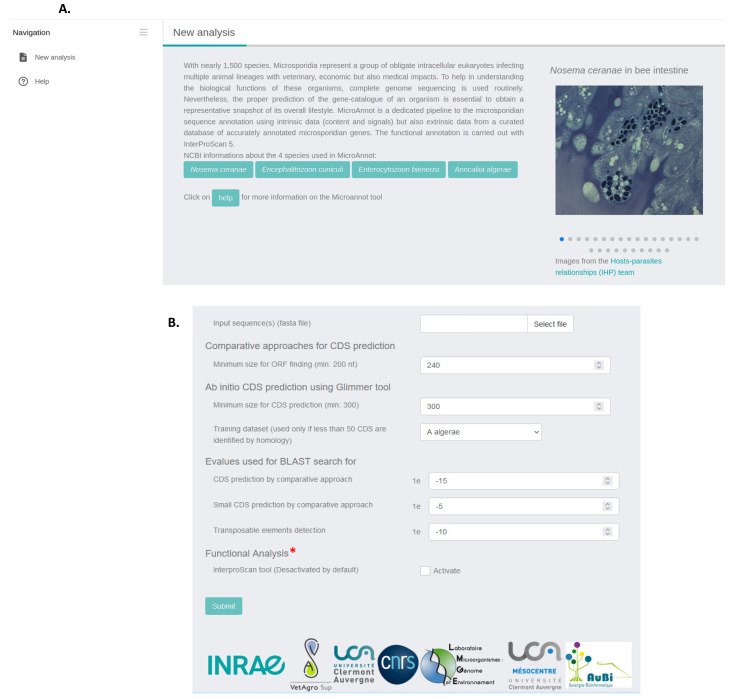
Microannot interface. (**A**) Homepage with the pipeline description. By clicking on help, the user is redirected to MicroAnnot scheme. (**B**) The analysis section is also available on the homepage. Files containing sequence(s) to be annotated in FASTA format must be loaded onto the application using the select file button. Then, the user can select the minimal size of ORF finding and the parameters used for Glimmer CDS prediction (minimal CDS size and training data set if less than 50 CDS are identified by homology). Here, default parameters are presented for each section. Functional annotation with InterProScan is disabled by default but can be activated by checking the activate box under functional annotation (*).

**Figure 2 ijms-25-00880-f002:**
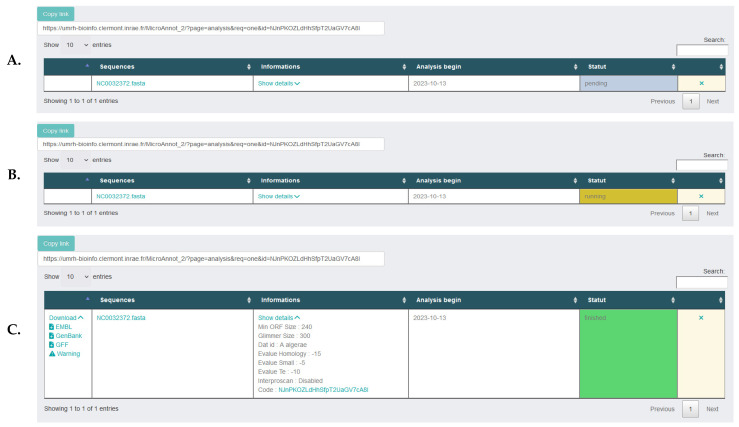
Web interface seen by the user after job submission. (**A**) pending, (**B**) running. (**C**) When the analysis is finished, different file formats (EMBL, GenBank or GFF) are available for download as well as a warning text file grouping uncertain annotations that must be manually validated. At each step, the sequence name (fasta file) is conserved, along with all the analysis details in the information column.

**Figure 3 ijms-25-00880-f003:**
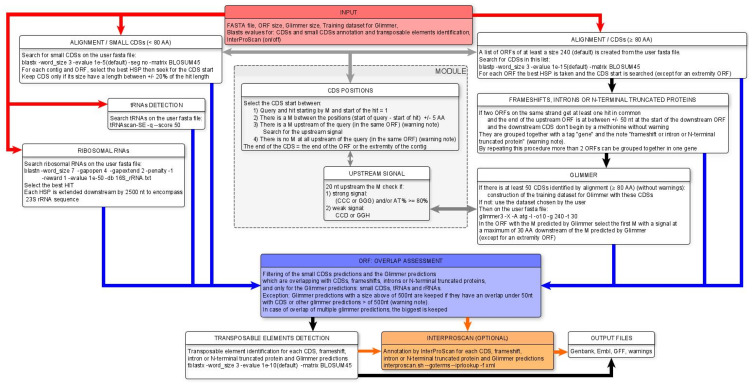
An overview of the workflow supported by the MicroAnnot pipeline. The different modules, using different bioinformatics programs and databanks, are displayed. A detailed description of each panel and module is provided in the text. Functional analysis by InterProScan, in orange (optional), needs to be selected in the first page of the interface (Figure 1B).

**Table 1 ijms-25-00880-t001:** Evaluation of annotation results performed by MicroAnnot on four well-annotated microsporidian genomes classified into three categories. (i) Annotation corrections proposed by MicroAnnot in comparison with annotation errors identified manually during different studies (light grey). Numbers in brackets correspond to the annotation corrections detected in this study by MicroAnnot; (ii) Additional annotation corrections identified with MicroAnnot (grey). (iii) Additional annotation errors obtained with MicroAnnot (dark grey). The number in brackets indicates the number of warnings encouraging authors to check these annotations. Detailed information can be found in Appendix A. TIS: Translational Initiation Site; TEs: Transposable Elements.

		*E. cuniculi*	*E. intestinalis*	*N. ceranae*	*E. bieneusi*
Annotation corrections	Falsely predicted TIS	445 (299: 67.2%)	199 (185: 93%)	308 (232: 75.3)	239 (171: 71.7%)
	Falsely predicted gene	11 (0: 100%)	8 (0: 100%)	76 (18: 76.3%)	168 (4: 97.6%)
	Newly predicted introns	3 (3: 100%)	19 (12: 63.2%)	4 (3: 75%)	-
	Unpredicted genes	142 (128: 90.1%)	115 (108: 93.9%)	292 (250: 86.5%)	70 (66: 94.3%)
	Unpredicted frameshift or sequencing error or pseudogene	-	11(10: 90.9%)	121 (74: 61.1%)	43 (17: 39.5%)
Additional annotation corrections	Corrected TIS	77	18	15	25
	Falsely predicted gene	12	-	3	8
	Predicted introns	-	-	1	-
	Unpredicted genes	23	-	296	15
	Unpredicted frameshift or sequencing error or pseudogene	-	1	-	-
	TEs predicted as CDS	-	-	42	-
	Predicted TEs	-	-	643	-
Additional annotation errors	Mispredicted TIS (warning)	94	32	71 (49)	72 (29)
	Falsely predicted gene (warning)	1	1	12 (27)	25
	Mispredicted intron	5	0	-	-
	Unpredicted genes	47	4	32	41
	Bad predicted Tes	0	1	-	-

**Table 2 ijms-25-00880-t002:** Comparisons of MicroAnnot performances with previous annotations and Funannotate pipeline. The specificity (Sp), the sensitivity (Sn) and TIS prediction of the genes are defined as: Sp = TP(TP + FP), Sn = TP(TP + FN), and TCP = TP(TP + FPT). TP: True positives, FN: false negatives, FP: false positives, and FPT: falsely predicted TIS. * True gene numbers obtained after compilation of all the reannotation studies [12,13,18] and present data are corrected by MicroAnnot annotation. ** First annotation references [10,11,38,39] for *E. bieneusi*, *N. ceranae*, *E. intestinalis* and *E. cuniculi* genomes, respectively. ^!^: Value, taking into account TEs predicted as CDS. NP: Not Performed.

		*E. cuniculi*	*E. intestinalis*	*N. ceranae*	*E. bieneusi*
True gene numbers *		2151	1940	2605	1770
Specificity (Sp)	1st annotation **	98.9%	99.6%	95.6%	91%
	MicroAnnot	99.9%	99.9%	98.9%	98.4%
	funannotate	NP	99.7%	96.5% (84.8% ^!^)	99.4%
Sensibility (Sn)	1st annotation **	92.9%	94.4%	81.3%	95%
	MicroAnnot	95.2%	99.4%	97.2%	97.1%
	funannotate	NP	86.4%	82.8%	82.8%
TIS correctly predicted (TCP)	1st annotation **	80.5%	90%	87.3%	86.8%
	MicroAnnot	90%	97.8%	94%	92.4%
	funannotate	NP	91.0%	89%	84.3%

## Data Availability

Data is contained within the article and Appendix A.

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
