# Peer review of "MicroAnnot: A Dedicated Workflow for Accurate Microsporidian Genome Annotation"

_ijms, 2024, doi:10.3390/ijms25020880_

Round 1
Reviewer 1 Report
Comments and Suggestions for Authors
Major comments
Line 107-110; " to identify divergent or small orthologous sequences". This statement gives this reviewer the impression that the "MicroAnnot" can also be used to predict ortholog sequences. What if, the users want to find the orthologous sequences against species other than those listed in the current tool? In the introduction, the authors also need to talk about the existing resources for gene prediction and annotation specific to (and/or widely used by the microsporidian community) the microsporidia genome. The 'MicroAnnot" has been designed and developed specifically for the microsporidian community, therefore, this is really important for the audience (with no microsporidia background) to know about these resources. Please evaluate and compare "MetaAnnot" with other tools (widely used by the microsporidian community) in terms of features and functionality. The authors could consider presenting the comparison in a table. At present, MicroAnnot provides analysis against four well-annotated genomes i.e., " Encephalitozoon cuniculi, Nosema ceranae, Enterocytozoon bieneusi, and Anncaliia algerae" only. What if, the users are also interested in annotating a newly sequenced microsporidia genome other than the above-mentioned four well-annotated genomes? The authors could consider setting up a GitHub repo for providing the scripts and workflow of the "MicroAnnot". This will be helpful for the Microsporidia community to set and use the workflow offline as well. Minor comments Line 31: Remove "Thanks to its web interface" Line 38: Please rephrase the sentence "Microsporidia are vastly represented in aquatic environments and food webs". It is not clear how Microsporidia are represented. Introduction: In the first paragraph, please provide the global statistics of the infections caused by Microsporidia species. Line 49-52; Cite the references to support the statement. Line 88-89; "Computational genome annotation ..... in computational biology"- Remove the sentence and rephrase the following sentence accordingly. Line 123; Cite references for GENBANK, EMBL, GFF The text in all the figures is impossible to read at least for this reviewer, please, provide high-resolution images for all of them. Figure 1; legend: Line 171: "Sequences to annotate (fasta files) have to be downloaded from user files." - which user files? Line 175; "can be activated by checking (*)." should be "can be activated by checking the activate box under functional annotation. Line 180; "can be picked up" should be "can be downloaded" Line 251; Provide the name of the database used in "a microsporidian-specific TE database". Line 253; "the translated CDS sequences" - what are these sequences? Is it an in-house database mentioned in line 518? line 255: "to the user In" - 'In' should be 'in'. Line 360-361; provide references to support the statement. Line 518: "a specific protein database was built" should be "a specific in-house protein database was built" or "a specific protein database was built in-house"Author Response
Line 107-110; " to identify divergent or small orthologous sequences". This statement gives this reviewer the impression that the "MicroAnnot" can also be used to predict ortholog sequences. What if, the users want to find the orthologous sequences against species other than those listed in the current tool?
The MicroAnnot tool successively uses two complementary approaches implemented for gene identification. The first, based on the exploitation of extrinsic data, is based on the search for orthologous sequences. To carry out this first step, we used annotation data from 4 genomes whose annotation had been manually curated. This first annotation step also provides high-quality data sets for the second annotation step based on intrinsic data using the Glimmer tool. As this comparative approach does not allow identification of all the genes, particularly those forming part of the microsporidian pan-genome, the second one ensures prediction of additional genes that are not present in the genomes of the 4 species (de novo prediction). This information about intrinsic approach is given in the lines 119-122: “Secondly, as this comparative approach does not allow identification of all the genes, particularly those forming part of the microsporidian pan-genome, these newly predicted genes were used as training set for the Glimmer tool [32]. Thus, an ab initio sequence annotation was carried out and the potential translation initiation codons for the newly identified sequences were once again validated using transcriptional signals.”
In the future, the high-quality annotations of additional genomes, will be implemented in the MicroAnnot tool, providing new microsporidian genes that will be used for the extrinsic approach, proved to be the most effective in terms of specificity, sensitivity and correct prediction of translation initiation codons. These perspectives have been included in the discussion part (lines 458 to 464).
In the introduction, the authors also need to talk about the existing resources for gene prediction and annotation specific to (and/or widely used by the microsporidian community) the microsporidia genome. The 'MicroAnnot" has been designed and developed specifically for the microsporidian community, therefore, this is really important for the audience (with no microsporidia background) to know about these resources. Please evaluate and compare "MetaAnnot" with other tools (widely used by the microsporidian community) in terms of features and functionality. The authors could consider presenting the comparison in a table.
As indicated in the introduction part, “microsporidian genomes have been annotated using ab initio protein predictions that were based primarily on the detection of Open Reading Frames (ORF) using various generalist software such as GeneMarkES, Augustus [25,26] or Glimmer”. Finally, the contribution of MicroAnnot compared with some of these tools was made by comparing the annotation results with the first annotation carried out for all tested genomes. Indeed, the first annotation for the species was carried out with the Glimmer tool (N ceranae and E. cuniculi), a blast approach (E intestinalis) and fungene tool (E. bieneusi). In order to not bias MicroAnnot annotation results, it should also be noticed that for the genomes of E. cuniculi, N. ceranae and E. bieneusi, the annotations were carried out by selecting only those genomes not corresponding to the genome to be annotated. This information is given in the materials and methods section, but for greater clarity is now included at the beginning of the paragraph 2.3 MicroAnnot Validation. " Annotation of E. cuniculi, N. ceranae and E. bieneusi genomes were carried out by selecting in MicroAnnot only the proteomes not corresponding to those of the genomes to be annotated. For E. intestinalis, all the MicroAnnot proteomes were used.”.
As suggested, we have also realized a comparison between MicroAnnot and Funannotate recently used to perform annotation of the genome of Hamiltosporidium tvaerminnensis (Angst et al. 2023). This pipeline exploits numerous software tools to deliver a consensus annotation result. In our case, AUGUSTUS annotation was performed using the training species model of E. cuniculi. GeneMArk-ES was used in self-training mode. Finally, SNAP and GlimmerHMM tools were trained with BUSCO microsporidia database which encompasses 600 genes (microsporidia_odb10 dataset, creation date: 2020-08-05). These details about Funannotate configuration were included in the Material and methods part. In addition, in order to present the Funannotate results, an additional supplementary table was generated (Table S5). Data on sensitivity, specificity and ability to detect TIS defined from Funannotate prediction were added to table 2.
A paragraph describing these results has also been included in the 2.3 MicroAnnot Validation part “To validate MicroAnnot relevance and robustness, an analysis was carried out using the Funannotate tool, a pipeline including numerous software dedicated to ensure gene prediction in both fungi and eukaryote genomes. Although this tool presents a high specificity with values close to those obtained with MicroAnnot it turns out to be less effective in predicting the entire gene repertoire of the genomes studied but also to identify correct TISs (Tables 2, and Table S5).”
These results were also discussed “To strengthen the relevance of the MicroAnnot which considers the specific characteristics of microsporidian genomes, a comparative analysis with the funannotate pipeline was performed. This tool gibes highly sensitivity but presents limits in ensuring the prediction of the complete gene repertoire and more especially small ones. Prediction of translation initiation codons is also less efficient. To improve these predictions, the tool can integrate RNA-seq data, and in this case proves to be particularly effective on the previous two points, but this requires RNA-seq data. The software also led to the prediction of numerous false introns due to the use of AUGUSTUS and GeneMarkHMM sofware which have been optimized to ensure their prediction. As for the prediction of all genes, and correct TIS, this tool also requires RNA-seq data while the MicroAnnot tool does not.
- The authors could consider setting up a GitHub repo for providing the scripts and workflow of the "MicroAnnot". This will be helpful for the Microsporidia community to set and use the workflow offline as well.
As suggested by the reviewer a GitHub was created (https://github.com/JeremyTournayre/MicroAnnot) and the web link have been added in the abstract as in the Material and methods part.
Minor comments
Line 31: Remove "Thanks to its web interface".
This part of the sentence was removed. The new sentence is: “MicroAnnot is available to the community at https://microannot.org and its code source at https://github.com/JeremyTournayre/MicroAnnot”
Line 38: Please rephrase the sentence "Microsporidia are vastly represented in aquatic environments and food webs". It is not clear how Microsporidia are represented.
As suggested the sentence has been modified such as follows: “Microsporidia can be highly represented in some aquatic environments and thus play a role in food webs [3–5]. Thus, three additional references were cited in the manuscript:
Chauvet, M.; Debroas, D.; Moné, A.; Dubuffet, A.; Lepère, C. Temporal Variations of Microsporidia Diversity and Discovery of New Host-Parasite Interactions in a Lake Ecosystem. Environ Microbiol 2022, 24, 1672–1686, doi:10.1111/1462-2920.15950.
Stentiford, G.D.; Becnel, -J. J.; Weiss, L.M.; Keeling, P.J.; Didier, E.S.; Williams, B. -a. P.; Bjornson, S.; Kent, M.-L.; Freeman, M.A.; Brown, M.J.F.; et al. Microsporidia - Emergent Pathogens in the Global Food Chain. Trends Parasitol 2016, 32, 336–348, doi:10.1016/j.pt.2015.12.004.
Ruan, Y.; Xu, X.; He, Q.; Li, L.; Guo, J.; Bao, J.; Pan, G.; Li, T.; Zhou, Z. The Largest Meta-Analysis on the Global Prevalence of Microsporidia in Mammals, Avian and Water Provides Insights into the Epidemic Features of These Ubiquitous Pathogens. Parasit Vectors 2021, 14, 186, doi:10.1186/s13071-021-04700-x.).
Introduction: In the first paragraph, please provide the global statistics of the infections caused by Microsporidia species.
As suggested, we added information about global statistics on the prevalence of microsporidia in the text : “ Numerous species can also be found infecting animals of veterinary and economic importance with prevalence rates reaching 15 % in bovines and 22 % in ovis [4,5]. In addition, some microsporidian species may be involved in human diseases with 17 species belonging to 10 genera [6] that have been described as leading to severe syndromes predominantly in immunocompromised patients [6]. This includes acquired immunodeficiency syndrome (AIDS), in whom the prevalence of microsporidia can be as high as 11.2 % [5], but also patients that have undergone organ transplants and been treated with immunosuppressive drugs with a prevalence of up to 13.2% [5–9].
Line 49-52; Cite the references to support the statement.
As suggested, three references to support the statement have been cited.
Williams, B.A.P.; Williams, T.A.; Trew, J. Comparative Genomics of Microsporidia. Exp Suppl 2022, 114, 43–69, doi:10.1007/978-3-030-93306-7_2.
Jespersen, N.; Monrroy, L.; Barandun, J. Impact of Genome Reduction in Microsporidia. Exp Suppl 2022, 114, 1–42, doi:10.1007/978-3-030-93306-7_1.
Corradi, N.; Slamovits, C.H. The Intriguing Nature of Microsporidian Genomes. Briefings in Functional Genomics 2011, 10, 115–124, doi:10.1093/bfgp/elq032.
Line 88-89; "Computational genome annotation ..... in computational biology"- Remove the sentence and rephrase the following sentence accordingly.
We removed the sentence and the following sentence has been rephrase “However, the microsporidian genome features previously described turn out to be the pitfall of classical computational methods [27] for producing accurate prediction of complete gene repertoires.”
Line 123; Cite references for GENBANK, EMBL, GFF.
These references were added in the text as in the reference list.
Clark, K.; Karsch-Mizrachi, I.; Lipman, D.J.; Ostell, J.; Sayers, E.W. GenBank. Nucleic Acids Res 2016, 44, D67–D72, doi:10.1093/nar/gkv1276.
Kanz, C.; Aldebert, P.; Althorpe, N.; Baker, W.; Baldwin, A.; Bates, K.; Browne, P.; van den Broek, A.; Castro, M.; Cochrane, G.; et al. The EMBL Nucleotide Sequence Database. Nucleic Acids Res 2005, 33, D29–D33, doi:10.1093/nar/gki098.
Annotating Genomes with GFF3 or GTF Files Available online: https://www.ncbi.nlm.nih.gov/genbank/genomes_gff/ (accessed on 20 December 2023).
The text in all the figures is impossible to read at least for this reviewer, please, provide high-resolution images for all of them.
High resolution figures have been included in individual files
Figure 1; legend: Line 171: "Sequences to annotate (fasta files) have to be downloaded from user files." - which user files?
The user file corresponds include the genomic sequence to be annotate by MicroAnnot. To clarify this part, the sentence was modified “File containing sequence(s) to annotate in FASTA format must be loaded onto the application using the select file button.”
Line 175; "can be activated by checking (*)." should be "can be activated by checking the activate box under functional annotation.
The suggested change was made in the legend of figure 1.
Line 180; "can be picked up" should be "can be downloaded".
As suggested the sentence was modified as follows: “Once the job is finished, several compressed archives containing GENBANK, EMBL and GFF formats can be directly downloaded (Figure 2C)”.
Line 251; Provide the name of the database used in "a microsporidian-specific TE database".
The TE database is an in-house database constructed using numerous previous works on microsporidian TE. As described in the Materials and Methods part in the databases section, as TE are very divergent in microsporidia species, we collected 1167 nucleic sequences (consensus or specific sequences) from previously published data (Ndikumana et al., 2017; Parisot et al., 2014; Peyretaillade et al., 2012; Song et al., 2020). This TE list was completed by TE sequences recently identified thanks to the complete chromosome assembly of the A. algerae genome using PacBio Hifi sequencing technology (unpublished data).
Line 253; "the translated CDS sequences" - what are these sequences? Is it an in-house database mentioned in line 518?
We apologize. These sequences were obtained by translation of MicroAnnot predicted CDSs and were then used as query for the InterProScan analysis. The sentence was therefore modified “Finally, a functional annotation of all translated CDSs predicted by MicroAnnot is carried out using InterProScan”
line 255: "to the user In" - 'In' should be 'in'.
the modification has been carried out in the text.
Line 360-361; provide references to support the statement.
Two references were added to support the statement.
Line 518: "a specific protein database was built" should be "a specific in-house protein database was built" or "a specific protein database was built in-house". 7
As suggested the sentence was modified as follows “a specific in-house protein database was built”.

Reviewer 2 Report
Comments and Suggestions for Authors
The manuscript entitled “MicroAnnot: a dedicated workflow for accurate microsporidian genome annotation ” describes a workflow and web interface for microsporidian sequence annotation.
Due to the development and wide application of the NGS technique in biological research, there is an urgent need for software for annotating the obtained genomic sequences. This manuscript provided such software. It is worth emphasising that MicroAnnot not only predict genes but also performs non-coding features annotation, detects sequencing errors and predicts putative transcription factors binding sites. The obtained results are reliable, clearly presented and interesting for readers. The manuscript is scientifically sound. Ethical issues do not raise concern.
Author Response
Thank you for your remarks about our manuscript.
Best regards
Round 2
Reviewer 1 Report
Comments and Suggestions for Authors
Thank you for answering my comments. I recommend the acceptance of the manuscript. However, there are few minor changes that I trust that the authors will include in the final proof before publication. Please see below:
1. Lines 32, 453; “code source” should be “source code”.
2. Lines 41-44; Please rephrase the sentence. This sentence is very confusing especially the line “belonging to 10 genera GenBank, EMBL, and GFF formats”.
3. Line 393; Cite the reference for “funannotate pipeline”.
4. Line 394; “gibes” should be “gives”
Author Response
Lines 32, 453; “code source” should be “source code”.
the modification has been done.
Lines 41-44; Please rephrase the sentence. This sentence is very confusing especially the line “belonging to 10 genera GenBank, EMBL, and GFF formats”.
we apologize because the error was due to an incorrect copy and paste. the sentence has been corrected as follows “In addition, some microsporidian species may be involved in human diseases with 17 species belonging to 10 genera that have been described as leading to severe syndromes predominantly in immunocompromised patient [6].”
Line 393; Cite the reference for “funannotate pipeline”.
Two citations have been added for the funannotate pipeline.
Line 394; “gibes” should be “gives”
the modification has been done.
